# Predicting Future Geographic Hotspots of Potentially Preventable Hospitalisations Using All Subset Model Selection and Repeated K-Fold Cross-Validation

**DOI:** 10.3390/ijerph181910253

**Published:** 2021-09-29

**Authors:** Matthew Tuson, Berwin Turlach, Kevin Murray, Mei Ruu Kok, Alistair Vickery, David Whyatt

**Affiliations:** 1Department of Mathematics and Statistics, University of Western Australia, Perth 6009, Australia; berwin.turlach@uwa.edu.au; 2School of Population and Global Health, University of Western Australia, Perth 6009, Australia; kevin.murray@uwa.edu.au (K.M.); meiruu.kok@uwa.edu.au (M.R.K.); 3Medical School, University of Western Australia, Perth 6009, Australia; alistair.vickery@uwa.edu.au (A.V.); david.whyatt@uwa.edu.au (D.W.)

**Keywords:** all subset model selection, place-based health service interventions, potentially preventable hospitalisations, repeated k-fold cross-validation, future hotspot prediction

## Abstract

Long-term future prediction of geographic areas with high rates of potentially preventable hospitalisations (PPHs) among residents, or “hotspots”, is critical to ensure the effective location of place-based health service interventions. This is because such interventions are typically expensive and take time to develop, implement, and take effect, and hotspots often regress to the mean. Using spatially aggregated, longitudinal administrative health data, we introduce a method to make such predictions. The proposed method combines all subset model selection with a novel formulation of repeated k-fold cross-validation in developing optimal models. We illustrate its application predicting three-year future hotspots for four PPHs in an Australian context: type II diabetes mellitus, heart failure, chronic obstructive pulmonary disease, and “high risk foot”. In these examples, optimal models are selected through maximising positive predictive value while maintaining sensitivity above a user-specified minimum threshold. We compare the model’s performance to that of two alternative methods commonly used in practice, i.e., prediction of future hotspots based on either: (i) current hotspots, or (ii) past persistent hotspots. In doing so, we demonstrate favourable performance of our method, including with respect to its ability to flexibly optimise various different metrics. Accordingly, we suggest that our method might effectively be used to assist health planners predict excess future demand of health services and prioritise placement of interventions. Furthermore, it could be used to predict future hotspots of non-health events, e.g., in criminology.

## 1. Introduction

In an environment with limited healthcare resources, it is essential to be able to accurately identify populations with excess burden of disease, in order to avoid inequality and effectively target interventions. Health service utilisation is often used as an indicator of health inequality; in particular, potentially preventable hospitalisations (PPHs), or ambulatory care-sensitive conditions (ACSCs), are widely used as an indicator of patients’ access to, and the provision and effectiveness of, primary care services [1,2,3,4,5,6]. Such hospitalisations are characterised by being potentially avoidable, or preventable, through provision of non-hospital or ambulatory health services; high rates of PPHs may indicate poor functioning of the primary health care system or care inappropriately directed to hospitals [3,7,8].

Ideally, individuals who are most at risk should be targeted for intervention; accordingly, a number of studies have focused on predicting, and examining the characteristics of, PPHs amongst individuals [9,10,11,12,13,14,15,16]. However, in practice, information on the behaviour and risk factors of individuals is often limited or unavailable, making it difficult or even impossible to develop appropriate individualised interventions [7,17]. Therefore, as an alternative, geographic areas with higher-than-expected rates of PPHs among residents, or “hotspots”, might be examined [17,18], and geographically targeted interventions subsequently developed.

Studies: examining geographic variation in PPHs; identifying individual- and area-level factors associated with such variation, e.g., socioeconomics, unemployment rates, and regionality; and classifying *current* hotspots of PPHs, are common (e.g., see [4,8,9,19,20,21,22,23,24,25,26,27,28,29,30]). Furthermore, an implicit assumption underpinning such studies is that current PPH hotspots will be predictive of future hotspots, and, thus, that they represent reasonable priority areas for targeted interventions (e.g., see [31]). However, place-based health interventions are typically costly and take time to develop and implement, and over short time periods current hotspots often regress to the mean [7]. Furthermore, at least for chronic PPHs, the time delay between the onset of disease and the development of complications leading to hospitalisation means that interventions aimed at reducing PPHs may take years to have an effect [31,32,33]. Therefore, targeting such interventions to current hotspots may be inappropriate or inefficient. Despite this, atlases of current variation in the utilisation of healthcare, with or without the subsequent explicit classification of hotspots, are widely used to guide clinical service planning and for research prioritisation. Examples include the Australian Atlas of Healthcare Variation [34], the USA Dartmouth Atlas Project [35], and the UK Atlas of Variation in Healthcare [36].

To address this deficiency, and in order to effectively guide health policy planners, it is essential to be able to accurately predict PPH hotspots several years into the future. However, while some longitudinal studies have been undertaken (e.g., see [4,37,38,39,40]), to the best of our knowledge, only one previous study has explicitly predicted, and evaluated the prediction of, future PPH hotspots: Duckett and Griffiths (2016) used past periods of persistently high PPH rates to predict areas expected to exhibit correspondingly high rates in the future [7]. Several subsequent studies have employed this approach in Australia [39,41]. However, as we will show, relying on past persistent “hotness” to predict future hotspots results in inappropriate prioritization of positive predictive value (PPV) over other metrics that are critical to performance and planning, e.g., sensitivity. Consequently, the utility of the past persistent hotspots approach for guiding placement of long-term, place-based health interventions aimed at reducing rates of PPHs, is limited.

In this paper, we introduce a novel method to predict PPH hotspots multiple years into the future. We begin by describing the classification of geographic hotspots based on directly age-sex standardised rates, before outlining the proposed method and illustrating its application predicting three-year future PPH hotspots for four chronic conditions in Western Australia (WA): chronic obstructive pulmonary disease (COPD), high-risk foot (HRF), heart failure (HF), and type II diabetes mellitus (T2D). In doing so, we compare the performance of our method to that of the current and past persistent hotspots prediction approaches described above. We end with a general discussion, including acknowledgement of limitations and suggestions for future work.

## 2. Materials and Methods

### 2.1. Defining Hotspots Based on Age-Sex Standardized Rates

Our method will utilise data aggregated to the level of small geographic areas. Such areas are typically administrative boundaries, which exist in many countries, e.g., census-block groups (CBGs) in the US [42], middle layer super output areas (MSOAs) in the UK [43], and Statistical Areas Level 2 (SA2s) in Australia. In WA there are 250 SA2s, including 173 in metropolitan Perth; these have a mean population of approximately 10,000 residents [44]. Figure 1a,b show the distribution of SA2s across WA and metropolitan Perth, respectively.

We define geographic hotspots of PPHs by calculating a directly age-sex standardised rate (ASSR) for each area j, within each year l, as [45]:(1)$$ASSR_{jl}=\frac{1}{\sum_{i}w_{i}}\underset{i}{\sum }\frac{w_{i}O_{ijl}}{n_{ijl}}$$
where $O_{ijl}$ and $n_{ijl}$ are the PPH count and population size, respectively, in age-sex stratum i, area j, and year l, and $w_{i}$ is the standard population for age-sex stratum i (typically taken from a recent national census). Upper and lower confidence limits for ${\mathrm{ASSR}}_{jl}$ are calculated as:(2)$$ASSR_{jl,\;lower}=ASSR_{jl}+\sqrt{\frac{Var\left(ASSR_{jl}\right)}{Var\left(O_{l}\right)}}\;\left(O_{l,\;lower}-O_{l}\right)$$
(3)$$ASSR_{jl,\;upper}=ASSR_{jl}+\sqrt{\frac{Var\left(ASSR_{jl}\right)}{Var\left(O_{l}\right)}}\;\left(O_{l,\;upper}-O_{l}\right)$$
where $O_{l}$ is the PPH count across all areas in year l. Upper and lower confidence limits for $O_{l}$ are calculated using Byar’s approximation [45]:(4)$$O_{l,\;lower}=O_{l}\times {\left(1-\frac{1}{9O_{l}}-\frac{z}{3\sqrt{O_{l}}}\right)}^{3}$$
(5)$$O_{l,\;upper}=\left(O_{l}+1\right)\times {\left(1-\frac{1}{9\left(O_{l}+1\right)}-\frac{z}{3\sqrt{O_{l}+1}}\right)}^{3}$$
where z is the normal quantile corresponding to the desired confidence level. The terms $Var\left({ASSR}_{jl}\right)$ and $Var\left(O_{l}\right)$ in Equations (2) and (3) are calculated as:(6)$$Var\left(ASSR_{jl}\right)=\frac{1}{{\left(\sum_{i}w_{i}\right)}^{2}}\underset{i}{\sum }\frac{w_{i}{}^{2}O_{ijl}}{n_{ijl}{}^{2}}$$
(7)$$Var\left(O_{l}\right)=\underset{i}{\sum }O_{il}$$

Additionally, in each year l, the ASSR of PPHs calculated using data from all areas (hereafter the “all-area” ASSR of PPHs) is calculated using Equation (1) as:(8)$$ASSR_{l}=\frac{1}{\sum_{i}w_{i}}\underset{i}{\sum }\frac{w_{i}O_{il}}{n_{il}}$$
where $O_{il}$ and $n_{il}$ are the all-area PPH count and population size, respectively, in age-sex stratum i.

Thus, in each year l, PPH hotspots are defined to be those areas j for which ${ASSR}_{jl,\;lower}>{ASSR}_{l}$, i.e., those areas with ASSRs of PPHs statistically significantly higher than the all-area ASSR of PPHs, at the specified confidence level.

### 2.2. Future Prediction of Hotspots

Our method is concerned with predicting hotspots m years into the future, where m is the estimated time required to develop and implement a proposed intervention (e.g., three years). To do this, we construct a dichotomous outcome variable that indicates, in each year, whether each area is a hotspot in years l+1,…,l+m. Corresponding area-level covariates are defined in year l, though the values of these covariates may depend on data from previous years. Due to the dichotomous nature of the outcome, logistic regression models are fitted. We note that alternative models for dichotomous outcomes, such as regression trees, may also be used; however, a comparison of such models is outside the scope of this study.

To identify optimal prediction models, multiple candidate models, comprising combinations of the available covariates, are compared using an all-subsets model selection approach [46]. In this process, a single optimal model is identified, which maximises a chosen performance metric, or multiple metrics, of interest. To compare models, repeated k-fold cross-validation (CV) [47,48,49] is undertaken.

Importantly, though the covariates are defined in a single year, as noted above, some may contain information from previous years (e.g., the number of past consecutive years classified as a hotspot). Therefore, to fully utilise the information contained in such covariates, it is necessary to maximise the available data prior to year l. With this in mind, in Section 2.3 we introduce an existing formulation of k-fold CV for longitudinal data, namely the rolling forecast window approach [50,51], and show how using this approach in our context results in the restriction of covariates utilising information from prior years. To mitigate this restriction, in Section 2.4 we develop an alternative formulation of k-fold CV for longitudinal data. Repeated k-fold CV is described in Section 2.5. Finally, in Section 2.6, we describe a calibration-implementation approach allowing the identification of optimal models based on multiple performance metrics of interest.

### 2.3. Cross-Validation for Longitudinal Data

The rolling forecast window approach is a commonly implemented CV method that takes into account the dependence of longitudinal observations. In this method, a window of width w time points (years, in the current discussion) is iteratively rolled forwards through the complete time period of width T>w years. A training/test evaluation is conducted in each iteration, resulting in a number of values, equal to the number of iterations conducted, being obtained for a performance metric of interest (e.g., sensitivity, PPV, or accuracy). These are combined (usually by averaging) to calculate the CV statistic.

To implement the rolling forecast window approach in the current context, where a logistic regression model is fitted to a dichotomous outcome defined over a future period of width m years, we fit the model:(9)$$\log\left(\frac{\pi_{a}}{1-\pi_{a}}\right)=\beta_{0}+\beta_{1}x_{a1}+\ldots +\beta_{p}x_{ap}$$

To the training sample in each iteration. Here, $\pi_{a}$ is the probability of a positive outcome case for SA2 a, and $\beta_{0},\beta_{1},\ldots ,\beta_{p}$ are the estimated regression coefficients for p area-level covariates. A classification threshold is selected by dichotomising the model-predicted probabilities for the training dataset and optimising the metric of interest. Subsequently, the fitted model is applied to the test data, which is shifted forwards m years in time compared to the training sample (so that the training and test outcome periods do not overlap). This shift mimics the later use of observed data to predict in a future, unobserved time period. The model-predicted probabilities for the test dataset are dichotomised using the previously selected threshold; comparing these predictions to the observed values in the test dataset produces a validation estimate for the chosen performance metric. Iterating the entire process, by rolling the training and test datasets forward in time by one-year increments, results in multiple values for the performance metric of interest; these are combined to calculate the CV statistic.

For example, suppose T=10; let m=1; and suppose that C=5 comparisons are required to reliably estimate the CV statistic (as is assumed in five-fold CV). Thus, five iterations of the rolling forecast window approach are needed. To obtain these, the first training sample must comprise covariate data defined in year 4, with the outcome defined in year 5; the corresponding values for the first test sample are years 5 and 6. The second training sample comprises covariate data defined in year 5, with the outcome defined in year 6; the corresponding test sample values are years 6 and 7, and so on. Thus, in order to obtain the required five comparisons, the rolling window has a maximum width of w=4 years. To reiterate the point made previously, while the covariate data are defined in a single year, for example in year 4 in the first iteration of the above example, the values of some covariates may rely on data from previous years (e.g., years 1 to 3, in this case).

This example illustrates a trade-off that exists in implementing the rolling forecast window approach within the proposed logistic framework: given T, the maximum value of w is restricted by the values m and C, and vice versa. In general, a researcher is, thus, faced with a difficult scenario, since: (a) predicting a number of years into the future (m) is usually necessary/required; (b) a certain number of comparisons (C) are usually required in order to reliably calculate a CV statistic; and (c) restricting the width of the forecast window (w) results in substantial loss of information for predictors utilising past data.

### 2.4. A New Formulation of K-Fold Cross-Validation for Longitudinal Data

Therefore, to mitigate the truncation of predictors that utilise past data, we propose an alternative method to the rolling forecast window approach. This approach is outlined below in the context of the logistic framework introduced previously.

First, divide the available data into K folds, where the partition is based on groups of areas, then iterate for k=1,…,K:

1.Leaving out data from fold k, fit the model in Equation (9) to the training dataset comprising K−1 folds, and select a classification threshold;2.Apply the fitted model and the selected threshold to the test dataset, which comprises data from the kth fold, shifted forwards m years in time. Compare the predictions to the observed values to obtain a value for the performance metric of interest.

The selection of a classification threshold in step i. above is based on optimisation of the performance metric of interest. This is unchanged from Section 2.3. The layout of the training and test datasets is also unchanged, except that, rather than iteratively rolling the training/test evaluations forwards in time, the observed data is divided only once into K folds. Thus, by construction, C=K comparisons are obtained, and, unlike with the rolling forecast window approach, the value of C is independent of the values of m and w. Returning to the example in Section 2.3, where m=1 and T=10, we see that, using our method, each of the training samples may comprise covariate data defined in year 8, with the outcome defined in year 9; the corresponding values for each test sample are years 9 and 10. Thus, using our method, the width of the forecast window has a maximum value of w=8 years, a substantial improvement on the maximum value of w=4 observed when using the rolling forecast window approach.

### 2.5. Repeated K-Fold Cross-Validation

Particularly when applying k-fold CV to small datasets, variability may arise due to the random partitioning of the available data into K folds. In the current context, where a single optimal model structure is selected by implementing the proposed CV procedure for each candidate model in an all-subset model selection design, this variability manifests in the selection of different model structures as “optimal” for different partitionings. Thus, we use repeated CV to stabilise the selection of a single optimal model structure. Repeated CV is commonly implemented to reduce the variability associated with single-run k-fold CV [48,49].

To illustrate, suppose that the performance metric of interest is accuracy. This is calculated for each model d by first “pooling” the predicted counts (true positives (TPs), false positives (FPs), false negatives (FNs), and true negatives (TNs)) for that model across CV folds and, subsequently, across CV repeats, according to the formulae:(10)$$TP_{d}=\underset{q}{\sum }\underset{k}{\sum }TP_{dqk}$$
(11)$$FP_{d}=\underset{q}{\sum }\underset{k}{\sum }FP_{dqk}$$
(12)$$FN_{d}=\underset{q}{\sum }\underset{k}{\sum }FN_{dqk}$$
(13)$$TN_{d}=\underset{q}{\sum }\underset{k}{\sum }TN_{dqk}$$
where $TP_{dqk}$, $FP_{dqk}$, $FN_{dqk}$, and $TN_{dqk}$ are the predicted counts observed when evaluating fitted model d on CV fold k, within CV repeat q. Additional discussion of the use of pooling to combine predicted counts in k-fold CV is given in the general discussion. Accuracy is then calculated as:(14)$$accuracy_{d}=\frac{TP_{d}+TN_{d}}{TP_{d}+FP_{d}+TN_{d}+FN_{d}}$$

### 2.6. Optimising Sensitivity and PPV in A Two-Step Calibration-Implementation Approach

Often, multiple performance metrics are of interest. For example, in the current context of predicting future PPH hotspots, we argue that both sensitivity (the proportion of outcome cases correctly predicted) and PPV (the proportion of correct predictions) are important. In such cases, in implementing the proposed method, an optimal model structure might be chosen which maximises one metric while maintaining the second metric above a pre-specified minimum threshold. Here, we describe a two-step calibration-implementation extension of the proposed method that achieves this. We present formulae for the maximisation of PPV while maintaining sensitivity above a minimum threshold; however, other combinations of metrics could also be examined using this approach.

First, in the calibration step, a small number of CV repeats are conducted over a grid of sensitivity thresholds between 0 and 0.9. At each threshold t, sensitivity and PPV are calculated for candidate models by pooling the predicted counts for each model according to the following formulae:(15)$$TP_{dt}=\underset{q}{\sum }\underset{k}{\sum }TP_{dqk}$$
(16)$$FP_{dt}=\underset{q}{\sum }\underset{k}{\sum }FP_{dqk}$$
(17)$$FN_{dt}=\underset{q}{\sum }\underset{k}{\sum }FN_{dqk}$$
(18)$$TN_{dt}=\underset{q}{\sum }\underset{k}{\sum }TN_{dqk}$$

Here, $TP_{dqk}$, $FP_{dqk}$, $FN_{dqk}$, and $TN_{dqk}$ are as defined in Equations (10)–(13), while additionally being subject to t. The pooled sensitivity and PPV for model d, corresponding to sensitivity threshold t, are then calculated as:(19)$$sensitivity_{dt}=\frac{TP_{dt}}{TP_{dt}+FN_{dt}}$$
(20)$$PPV_{dt}=\frac{TP_{dt}}{TP_{dt}+FP_{dt}}$$

By observing the PPV obtainable across the sensitivity grid, an “appropriate” minimum sensitivity threshold, $t^{*}$, may be selected for the condition being examined. In practice, the selection of $t^{*}$ might also be based on the requirements of a proposed intervention (i.e., it may be pre-selected). For example, if there is no minimum sensitivity required, then $t^{*}=0$ and an optimal model could be selected based on maximising PPV only, without using the calibration-implementation approach.

Following the calibration step, in the implementation step, numerous CV repeats are conducted at $t^{*}$ and an optimal model structure is selected. Preliminary work suggested that a large number of CV repeats would be required to stabilise the model selection process, compared to a relatively low number of repeats reported elsewhere [52,53]. Therefore, results based on 250 CV repeats are presented in the application examples.

### 2.7. Real-World Applications: Data and Methods

Hospital admissions for COPD, HRF, HF, and T2D, occurring in the (Australian) financial years (July to June) between 2002–2003 and 2014–2015 inclusive, were extracted from the Hospital Morbidity Data Collection, one of the extensive WA linked data collections [54]. Admissions were identified using diagnosis/procedure codes from the International Statistical Classification of Diseases and Related Health Problems, Tenth Revision, Australian Modification (ICD-10-AM) [55]. This modification is maintained to ensure that the ICD classification is current and appropriate for Australian clinical practice. Admissions for COPD, HF, and T2D were identified by the principal diagnosis codes J44, I50, and E11, respectively; the latter excluding admissions with a principal diagnosis code of E11.39 (“type II diabetes mellitus with other specified ophthalmic complication”). Admissions for HRF were identified based on codes enumerated in a 2014 Department of Health, Western Australia report [56]; specifically, codes identifying admissions for “Non-Traumatic Minor Amputation: Toe”; “Non-Traumatic Major Amputation: Above Knee”; “Non-Traumatic Major Amputation: Below Knee”; “Osteomyelitis of the foot/ankle”; “Diabetic foot”; “Peripheral Vascular Disease”; “Cellulitis of the Lower Limb” and “Charcot’s Foot”. Principal diagnosis codes identifying traumatic amputations were excluded.

The extracted admissions were aggregated by financial year and SA2 boundaries to form state-wide and metropolitan datasets (comprising 250 SA2s across WA and 173 metropolitan SA2s, respectively). This was done because substantial variation in health utilisation outcomes is often observed between regional and metropolitan WA, but sparse data in regional areas means that such areas frequently cannot be analysed alone. The admissions data were merged with population data interpolated and extrapolated around census population data from 2001, 2006, and 2011, obtained from the Australian Bureau of Statistics. Areas with a population size less than 1000 in any financial year (these predominantly comprised airports, national parks, and industrial areas) were excluded, leaving 149 metropolitan SA2s and 222 SA2s across WA.

For each condition examined, ASSRs of hospital admissions were calculated using Equation (1) for combinations of SA2 and year, with the standard population taken to be the Australia-wide population from the 2011 census. Then, hotspot SA2s were identified in each year as those with ASSRs of PPHs statistically significantly higher than the all-area ASSR (the latter calculated using Equation (8)). In strata with a small population size, aggregation across age groups is often recommended in order to avoid instability that may arise when calculating standardised rates; such aggregation has been recommended for groups with population denominators less than 30 [57]. An alternative approach was implemented here, where the observed admission counts and populations in age-sex strata with population sizes less than 30, for particular combinations of area j and year l, were substituted by the corresponding all-area admission counts and population sizes, respectively (i.e., substituting $O_{ijl}$ and $n_{ijl}$ by $\underset{j|i,l}{\sum }O_{ijl}$ and $\underset{j|i,l}{\sum }n_{ijl}$ in Equations (1), (6) and (7), for relevant age-sex strata i). In this way, the distribution of the population among age-sex strata in specific SA2s is retained, and no loss of information occurs as would be the case if aggregating across age groups in the ad hoc manner described above.

A binary outcome was constructed, where positive outcome cases were SA2s classified as hotspots in each of three (i.e., m=3) consecutive future years (i.e., between 2009–2010 and 2011–2012 in the training sample and between 2012–2013 and 2014–2015 in the test sample). Next, area-level candidate predictors in the year in which predictors were defined (l=2008–2009 for the training samples and l=2011–2012 for the test samples) were selected based primarily on past research. These included:the percentage of individuals identifying as Aboriginal and/or Torres Strait Islander (hereafter “Aboriginal”);the percentage of individuals aged 75 years or older (along with a quadratic term);the percentage of male individuals (centred around the mean within each financial year);rurality (metropolitan or regional);accessibility to emergency department (ED) and general practice (GP; in the US read “family practice”); andfour Socio-Economic Indices for Areas (SEIFAs): (i) the Index of Relative Socio-Economic Disadvantage (IRSD), (ii) the Index of Relative Socio-Economic Advantage and Disadvantage (IRSAD), (iii) the Index of Education and Occupation (IEO), and (iv) the Index of Economic Resources (IER) [58].

A percentile rank was constructed for each SEIFA index. Accessibility to ED was calculated as the population-weighted distance to the nearest ED (in kilometres) using information from the WA Emergency Department Data Collection [54]. Accessibility to GP was calculated using 2015 information from the WA Primary Health Alliance as either: (i) the population-weighted distance to the nearest GP clinic (in kilometres) or (ii) the GP accessibility index obtained from a gravity-based accessibility model, adjusted for the distance to each GP clinic, the population within each area, and the population competition between areas [59]. A percentile rank was constructed for the latter. The number of past consecutive years classified as a hotspot was also included as a predictor, along with interaction terms with each SEIFA percentile. Interactions between the weighted distance to nearest ED and each SEIFA percentile were also considered. The distance variables and the percentage of Aboriginal individuals were log-transformed to account for skewness, with an adjustment in the latter case to account for the presence of zeros [60]. This adjustment facilitates the approximate log-transformation of continuous predictors x, which have some zero values, according to the formula:(21)$$f\left(x\right)=\{\begin{array}{c} x\;\;\;\;\;\;\;\;\;\;\;\;\;\;\;\;\;\;\;\;\;\;\;\;\;\;\;\;\;\;\;if\;\;\left|x\right|\;\leq c\\ sign\left(x\right)\times c\times \left(1+\log\left[\frac{\left|x\right|}{c}\right]\right)\;\;\;\;\;\;\;if\;\;\left|x\right|\;\geq c\; \end{array}$$
where c is a scalar, which was set to 1.

For each condition, a group of candidate models was constructed using a modified all-subset model selection approach; included models consisted of all permutations of the candidate predictors except: permutations violating the marginality principle; those containing multiple SEIFA indices; and those containing both measures of GP accessibility. Since outcome cases were relatively uncommon for the four PPHs examined, models with more than five predictors were excluded to avoid potential problems arising due to data sparsity. This restriction was concordant with preliminary results showing no improvement in PPV for larger models. Separate state-wide and metropolitan groups were assembled comprising the remaining permutations; there were d=1146 candidate models in the state-wide group and d=726 candidate models in the metropolitan group.

For each condition, the method described in Section 2.4 was implemented. Candidate models were fitted using logistic regression with the Firth correction [61,62,63] to avoid problems that may arise due to data separation. Using the calibration-implementation procedure described in Section 2.6, five repeats of five-fold CV were conducted in the calibration step, and a different sensitivity threshold was selected for each condition. It is frequently recommended that between five and ten folds be used when implementing k-fold CV, though this choice remains an open area of research [64]; we used five-fold CV in order to balance the number of folds against (a) the number of areas in each fold, and (b) the increased computational load resulting from the use of more folds (e.g., 10) in combination with repeated k-fold CV and all subset model selection.

Subsequently, 250 CV repeats were conducted in the implementation step, and a single optimal model was selected for each condition. In each case, the structure of this model was then applied to the most recent data available (i.e., predictor information from 2011–2012, with the outcome defined between 2012–2013 and 2014–2015), to predict future hotspot SA2s up until 2018–2019.

## 3. Results

### 3.1. Results from Real-World Applications

In applying our method, outcome cases in the training samples (i.e., SA2s classified as hot between 2009–2010 and 2011–2012) were uncommon for all PPHs examined, ranging between 5 SA2s (2.3% of all SA2s; HF) and 16 SA2s (7.2%; COPD) in the state-wide training samples and between 4 SA2s (2.7%; T2D) and 11 SA2s (7.4%; COPD) in the metropolitan training samples (Table 1). In particular, the presence of just four outcome cases in the metropolitan dataset for T2D meant that we had to use four-fold CV, instead of five-fold CV, in analysing this dataset; this point is revisited in the Discussion.

Figure 2 shows the validation sensitivity and PPV of candidate state-wide HRF models across the grid of sensitivity thresholds between 0 and 0.9 (Figure 2a–f). This figure was constructed using data from the calibration step of the state-wide HRF model. At each threshold, the optimal model is the one with maximum PPV that also has sensitivity greater than the specified threshold. For thresholds 0 and 0.2, there are multiple optimal models, each with PPV of 1, but at other thresholds a single optimal model is identifiable. Two phenomena are illustrated that are characteristic of all PPHs examined: first, PPV of the optimal model was lower at higher sensitivity thresholds, and second: clusters of models were observed. The latter point is discussed in more detail later. The inverse relationship between PPV and sensitivity is illustrated further in Figure 3, where PPV from the optimal models for HRF (Figure 3a), COPD (Figure 3b), HF (Figure 3c), and T2D (Figure 3d), obtained using data from the calibration step for each model, is plotted across the sensitivity grid. Data for the optimal state-wide models are shown as solid lines with dashed lines indicating metropolitan data. Based on these data, a minimum sensitivity threshold of 0.5 was selected for the state-wide and metropolitan models for HRF, a threshold of 0.4 was selected for the corresponding COPD models, and a threshold of 0.3 was selected for the corresponding models for HF and T2D. These values are shown in Table 1.

In the implementation step, single optimal model structures were generally identified for each condition using a maximum of 50 CV repeats (though 250 CV repeats were conducted in each case). Thus, in general, increasing the number of CV repeats stabilised the selection of a single optimal model structure. However, in some cases, stabilisation did not occur even after pooling predictions across 250 CV repeats. Two contrasting examples of this phenomenon are presented in Figure 4, using data from the implementation step for the metropolitan COPD and state-wide T2D models. Figure 4a,b show the sensitivity and PPV for the “best” ten metropolitan COPD models, i.e., those which maintained the required sensitivity level of 0.4 and had the highest PPV, plotted against the number of CV repeats used (from 1 to 250). Corresponding data for the best ten state-wide models for T2D are shown in Figure 4c,d. While sensitivity for each of the COPD models was well above the minimum threshold of 0.4 (Figure 4a), some models had similar PPV (Figure 4b). By contrast, while one T2D model consistently had greater PPV than its counterparts (Figure 4d), its sensitivity fluctuated around the minimum threshold of 0.3 (Figure 4c). Both scenarios resulted in non-stabilisation of the choice of optimal model as additional CV repeats were conducted. Note that the latter phenomenon may not have arisen if only PPV was considered, since one model clearly had higher PPV than the others. The potential effects of selecting optimal models based on multiple performance metrics (as opposed to a single metric) is explored further in the Discussion.

As shown in Figure 2, clusters of models with similar PPV were sometimes observed at particular points in the grid of sensitivity thresholds. Models in such clusters typically comprised similar predictors: for example, Table 2 provides implementation-step sensitivity and PPV data for the ten best state-wide HRF models, along with mean odds ratios for the predictors in each of the ten models. The mean odds ratios were obtained by averaging estimated odds ratios from fitted models across multiple CV folds and CV repeats. The ten best state-wide models did not differ substantially: PPV ranged between 0.86 and 0.87 and some predictors appeared in several models. Positive associations were observed for the number of consecutive past years being classified as a hotspot and the percentage of Aboriginal individuals in an SA2, both of which appeared in all ten models, while negative associations were observed for the percentage of male individuals in an SA2, which appeared in nine of the ten models. Note that, while such observations may aid construction of an initial set of candidate predictors, the models described here were developed to maximise predictive performance rather than to examine association. Therefore, the odds ratios are not of primary importance and should be interpreted with caution [65,66].

Table 1 shows validation statistics for the optimal state-wide and metropolitan models for each condition, along with the number of SA2s predicted by each model. Uncertainty in each statistic is represented by the 2.5% and 97.5% quantiles from its distribution across the 250 CV repeats. The optimal state-wide models generally had greater PPV (ranging from 0.6; 95% confidence interval: 0.47–0.75 for COPD to 0.91; 0.63–1 for T2D) than the metropolitan models (ranging from 0.3; 0.14–0.38 for T2D to 0.77; 0.56–1 for HRF). This was possibly due to high variability between metropolitan and regional SA2s compared to relatively low variability between metropolitan SA2s. PPV for the metropolitan models for HRF (0.77; 0.56–1) and COPD (0.5; 0.38–0.64) exceeded that for HF (0.32; 0.22–0.5) and T2D (0.23; 0.14–0.38), possibly due to relatively low outcome counts for the latter two models, where a small number of false positives can heavily impact PPV [67].

Finally, Table 3 lists the predictors in the optimal models. Some predictors were important for multiple conditions; the number of past consecutive years classified as a hotspot was present in all state-wide models except for T2D and the percentage of Aboriginal individuals in an SA2 was present in all state-wide models, as well as the metropolitan models for HRF and T2D. Other predictors were less consistently present. These data are presented because they are likely to be informative in guiding the selection of a group of candidate predictors for certain conditions.

### 3.2. Comparison to Existing Methods

In this section, we compare the performance of our method to two approaches commonly used to predict future hotspots of PPHs, namely: (i) using current hotspots as a prediction rule (hereafter the “current hotspots” approach), and (ii) using past persistent hotspots as a prediction rule (hereafter the “past persistent hotspots” approach).

To give an example, Figure 5 shows the 22 hotspot SA2s for HRF in metropolitan Perth in 2011–2012 (light grey shading), and the subset of seven of these that were classified as hotspots for six consecutive years up until 2011–2012 (inclusive). These are the SA2s predicted to be future hotspots using the current and past persistent hotspots rules, respectively. Note that, while ten consecutive years were proposed for the past persistent hotspots rule by Duckett and Griffiths (2016), we used six years due to the relative scarcity of persistent hotspots for HF and T2D (this is possibly due to the relatively low number of SA2s in WA and metropolitan Perth, compared to cities and states in eastern Australia).

Of the 22 current hotspots, only six remained hotspots in 2014–2015 (i.e., in three years’ time). Thus, PPV for the current hotspots approach for HRF in metropolitan Perth was approximately 27%. This data is shown in Table 4 along with corresponding values for HRF, COPD, and HF. Corresponding estimates of sensitivity are also shown. Across the four conditions, PPV of no more than 27% was observed (HRF), while sensitivity ranged between 13% (T2D) and 55% (HRF). Corresponding data for the four conditions in all of WA are also shown; here, the highest PPV observed was 41% (HRF) while sensitivity ranged between 53% (T2D) and 68% (HRF).

Of the seven past persistent hotspots in 2011–2012 (Figure 5), four remained hotspots in 2014–2015 (dark grey shading); giving PPV of 57%. This data is similarly shown in Table 4 along with corresponding values for COPD, HF, and T2D, as well as corresponding estimates of sensitivity. PPV of no more than 57% was observed (HRF), while sensitivity ranged between 0% (HF and T2D) and 36% (HRF). Analogous data for the four conditions in all of WA showed PPV as high as 80% (HRF), with sensitivity ranging between 8% (HF) and 42% (HRF).

These results demonstrate that the current hotspots approach tends to give high sensitivity but relatively low PPV. Thus, though future hotspots are frequently identified, predictions based on this method are often incorrect. This reflects the fact that current hotspots frequently regress to the mean in the long-term. In contrast, the past persistent hotspots approach tends to give high PPV but relatively low sensitivity. Thus, while this method frequently predicts some future hotspots correctly, it often fails to identify others. Furthermore, in cases where there are no past persistent hotspots, this method cannot make any predictions, resulting in undefined PPV and zero sensitivity. This phenomenon was observed for the metropolitan HF dataset in the current examples.

## 4. Discussion

In this paper, we have developed a novel statistical method to predict future geographic hotspots of PPHs. This method incorporates all-subset model selection and a unique formulation of repeated k-fold CV for longitudinal data. Results from its application examining PPHs for four chronic conditions in an Australian context have illustrated its utility for accurately predicting future hotspots of PPHs and other health events, and its flexibility in maintaining required minimum performance criteria associated with proposed health interventions.

Although good predictive performance was generally observed across the conditions examined, relatively low PPV was observed in some cases. This suggests that predicting future persistent hotspots is inherently difficult for some conditions. However, regardless, we have shown that our method performs favourably compared to the two existing methods that are commonly used in practice, i.e., the current and past persistent hotspots approaches. Specifically, we have shown how current hotspots typically predict future hotspots with low PPV and high sensitivity, while past persistent hotspots typically predict future hotspots with low sensitivity and high PPV. Moreover, importantly, these characteristics are fixed; users are unable to adjust them to suit alternative requirements (i.e., of interventions). By comparison, our method allows users to optimise sensitivity; PPV; both sensitivity and PPV (as we have done); or any other metric(s) that might be of interest. A broad range of results are, thus, obtainable using our method, including, but not restricted to, similar values to those obtainable using the current and past persistent hotspots approaches.

While hotspot prediction models are often evaluated using a single metric (e.g., discrimination or calibration, see [68,69]), we have argued that both sensitivity and PPV are important in the context of planning and evaluating public health interventions. Costly interventions generally require high PPV, often to the detriment of sensitivity, while inexpensive interventions may sacrifice PPV to improve sensitivity, particularly if the cost of false positives is negligible. In general, a given optimisation rule should be tailored to the characteristics of the condition being examined and the feasibility of a proposed intervention. When examining multiple metrics, we recommend using the calibration-implementation approach we have presented. In our application, where PPV is maximised while maintaining sensitivity above a minimum level, the calibration step allows users to first assess whether a desired level of PPV is attainable at a sensitivity appropriate to a proposed intervention, before proceeding to the implementation step.

The choice of a minimum level of sensitivity is one of several inputs to our system that can be manipulated by users to suit their needs. Other choices include: the length of the outcome time period, which should correspond to the estimated time required to develop and implement a planned intervention (we have used three years as an example); which candidate predictors to consider; the maximum number of predictors to include in each model; and which performance metrics to optimise (again, these should be chosen based on a cost-benefit analysis of a proposed interventions). It is easy to envisage a computer application that, in implementing our method, allows users to easily manipulate these constraints to best suit their needs. The development of such an application represents an important area for future work.

Alternative methods to pooling may be used to calculate a CV statistic in k-fold CV. Usually, estimates are averaged across folds [49]. However, for datasets with a relatively low number of outcome cases, this might result in undefined estimates in some folds [70]. For example, sensitivity is undefined in folds with zero outcome cases, while PPV is undefined in folds with zero positive predictions. Such estimates may go undiscovered when buried within results that are aggregated over many folds; consequently, many researchers avoid drawing conclusions based on results that are derived using datasets with rare outcomes. However, it is important to consider the performance of modelling techniques such scenarios, particularly as they often occur in the medical domain [71]. In practice, undefined estimates are either accounted for by: (i) substituting zero, or (ii) exclusion. In small samples with rare outcomes, both of these approaches lead to biased estimates; by contrast, bias is minimised for some metrics (including sensitivity and PPV) when using pooling [71]. In our examples, pooling avoids the problem of undefined sensitivity since there is at least one outcome case in all samples. Furthermore, cases of undefined PPV are avoided by pooling across CV repeats.

We have examined population-based hotspots, as opposed to hotspots based on the prevalence of disease (prevalence-based hotspots), in order to emphasise interventions to reduce costs associated with hospitalisations. However, some studies have argued that population-based hotspots are less informative than prevalence-based hotspots since disease prevalence could vary between populations and between areas [72,73]. While further comment on this discussion is outside of the scope of this paper, we note that our method can be used to predict either type of hotspot, or both.

Applications of our method may be limited in some cases by time-dependent influences, e.g., changes in coding practices. For example, in our data, a change in coding practices for T2D in July 2010 resulted in far fewer admissions being coded using ICD-10-AM code “E11” following the change. The potential impact of such changes should be considered when applying our method. A second limitation is that, in extreme cases, there may be no persistent hotspots and, thus, no outcome cases. This was observed to a minor degree in the metropolitan model for T2D, where only four outcome cases were identified and, consequently, four-fold CV was used instead of five-fold CV. This limitation could potentially be overcome through utilizing models that do not require preliminary dichotomization of the outcome, e.g., those that directly model ASSRs of PPHs. However, the phenomenon is also partly attributable to the arbitrary nature of the areal boundaries used (SA2s); if a hotspot overlaps multiple SA2s, it may be diluted among them and, thus, go undiscovered. The impact of the particular choice of boundaries on analyses of areal data is described by the modifiable areal unit problem (MAUP) [74]. Future work should generalise our method to address the MAUP; specifically, its “scale” and “zonation” aspects, which describe, respectively, the dependence of a given analysis on the size and configuration of the chosen set of spatial boundaries. A recently proposed method that involves combining information across numerous zonations of fine-resolution data, in order to classify “zonation-independent” hotspots, could potentially be used for this purpose [75]. However, the choice of scale should also be carefully considered; this should be related to the scale of a proposed intervention [75]. To facilitate these endeavours, and in order that users have sufficient control over scale, as fine resolution data as possible should be obtained in the first instance; where fine-resolution data exist, but are unavailable, their custodians should be lobbied for access to those data based on a need to address the MAUP. Finally, our method might be improved through consideration of different sets of covariates, or utilization of dichotomous-outcome models other than logistic regression. All of these extensions can easily be incorporated into the framework we have developed.

## 5. Conclusions

In summary, we have presented a novel method to predict future geographic hotspots of PPHs. Characterised by its ability to maintain user-specified performance criteria associated with planned, place-based health interventions, the method is differentiated from existing methods to predict future PPH hotspots, namely current and past persistent hotspots approaches. In examining several real-world examples, we have demonstrated superior performance and flexibility of our method as compared to those alternatives. Consequently, we suggest that our method might usefully be used to assist health policy planners predict future demand and assess the potential benefits of geographically targeted health interventions. Furthermore, it could be used to predict future geographic hotspots of non-PPH health conditions and disease states, and of non-health-related events (e.g., in criminology). However, when using our method, policy makers and clinical planners should optimise it according to the characteristics of the cohorts, conditions, and proposed interventions under consideration.

## Figures and Tables

**Figure 1 ijerph-18-10253-f001:**
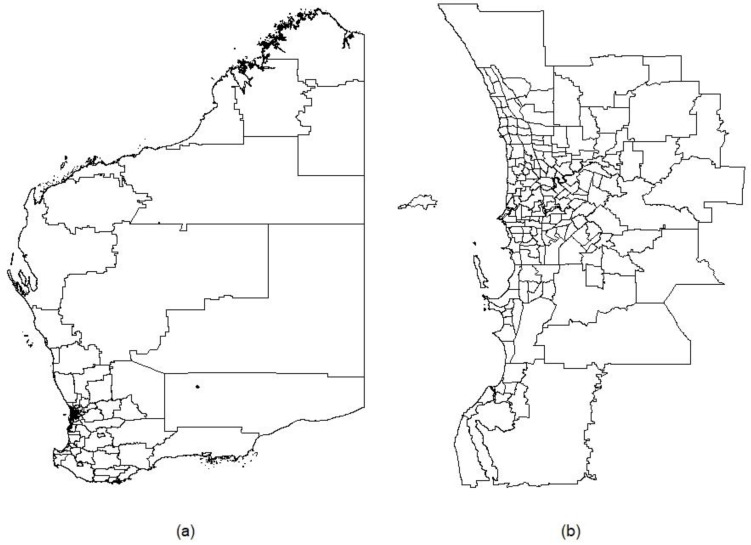
2016 Statistical Areas Level 2 (SA2) boundaries in (**a**) Western Australia and (**b**) metropolitan Perth.

**Figure 2 ijerph-18-10253-f002:**
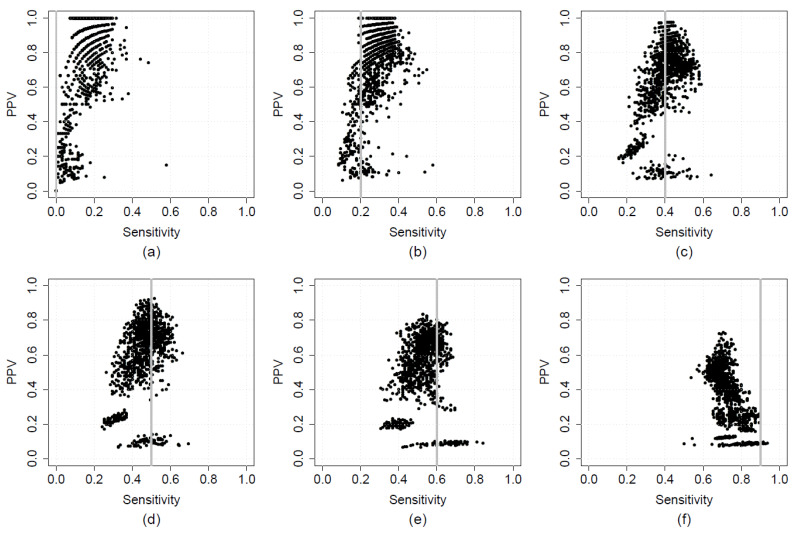
Validation positive predictive value (PPV) and sensitivity for candidate state-wide high-risk foot (HRF) models, plotted over a grid of minimum sensitivity thresholds from 0 to 0.9 (**a**–**f**). The minimum sensitivity thresholds are indicated by vertical grey lines. In each panel, black dots depict data for individual models.

**Figure 3 ijerph-18-10253-f003:**
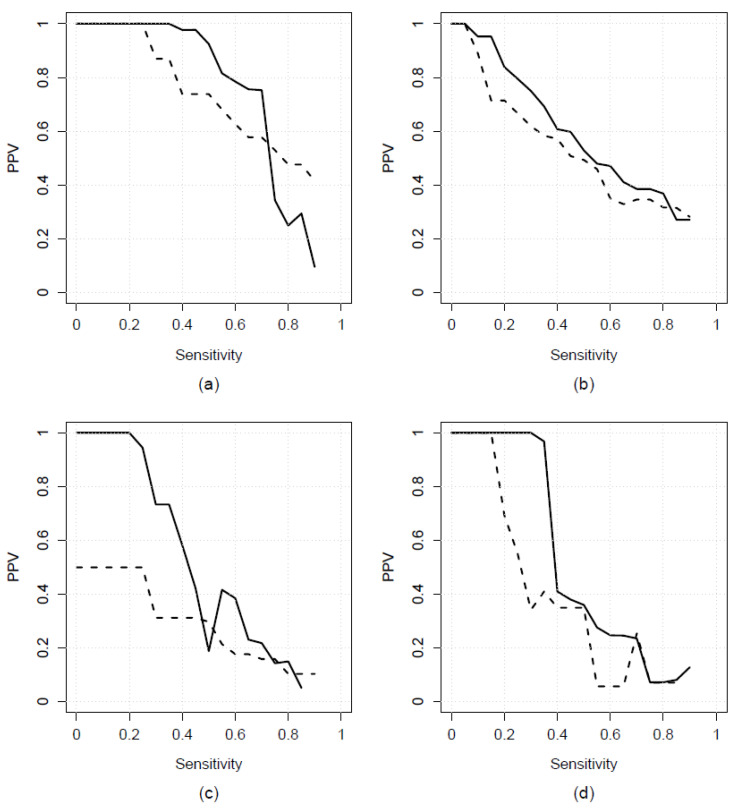
Validation positive predictive value (PPV) at various sensitivity thresholds for the optimal state-wide (solid) and metropolitan (dashed) models for: (**a**) high risk foot (HRF), (**b**) chronic obstructive pulmonary disease (COPD), (**c**) heart failure (HF), and (**d**) type II diabetes (T2D).

**Figure 4 ijerph-18-10253-f004:**
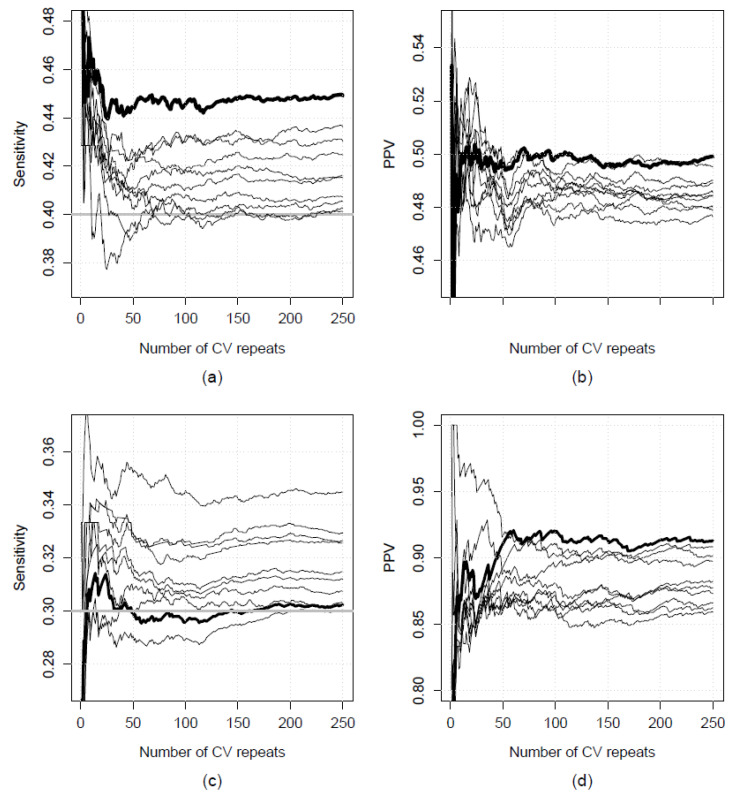
Validation sensitivity and positive predictive value (PPV) for: (**a**,**b**) the ten best metropolitan chronic obstructive pulmonary disease (COPD) models, and (**c**,**d**) the ten best state-wide models for type II diabetes (T2D). Heavier lines indicate the optimal models. The minimum sensitivity threshold is shown as a horizontal grey line in (**a**–**c**). CV: cross-validation.

**Figure 5 ijerph-18-10253-f005:**
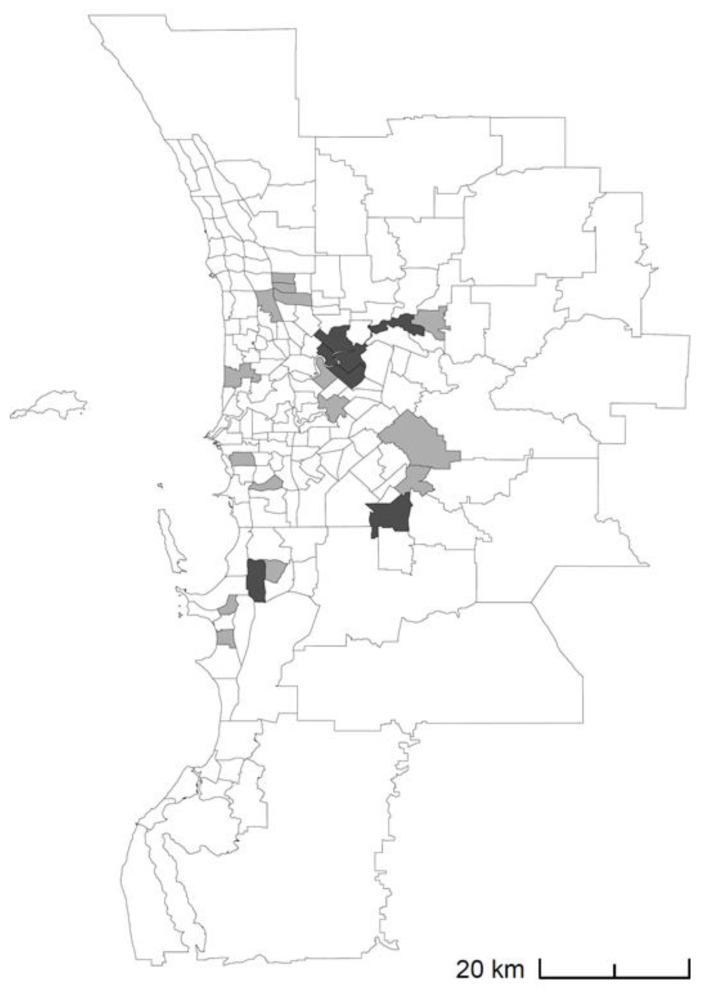
Hotspot SA2s for high-risk foot (HRF) in metropolitan Perth in 2011–2012 (all shaded areas), and the subset of seven of these that were classified as hotspots for six or more consecutive years up until 2011–2012 (inclusive; shaded dark grey).

**Table 1 ijerph-18-10253-t001:** Validation statistics for optimal state-wide and metropolitan models for high-risk foot (HRF), chronic obstructive pulmonary disease (COPD), heart failure (HF), and type II diabetes (T2D), with 95% quantile intervals. The number (N) and percentage (%) of outcome cases in each training sample; the sensitivity thresholds selected in the calibration step; and the number and percentage of predicted areas are also shown.

PPH	Validation Statistics (95% Quantile Interval)						
	N (%) Events ^1^	Sensitivity Threshold	Sensitivity	Specificity	PPV	NPV	N (%) Predicted ^1^
State-wide models							
**HRF**	14 (6.3)	0.5	0.531 (0.421–0.632)	0.993 (0.980–1.000)	0.872 (0.717–1.000)	0.958 (0.948–0.966)	11 (5)
**COPD**	16 (7.2)	0.4	0.412 (0.286–0.524)	0.972 (0.955–0.985)	0.602 (0.468–0.750)	0.941 (0.929–0.951)	17 (7.7)
**HF**	5 (2.3)	0.3	0.347 (0.250–0.481)	0.990 (0.976–1.000)	0.66 (0.444–1.000)	0.964 (0.958–0.971)	9 (4.1)
**T2D**	12 (5.4)	0.3	0.302 (0.200–0.333)	0.998 (0.986–1.000)	0.913 (0.625–1.000)	0.952 (0.945–0.954)	5 (2.3)
**Metropolitan models**							
**HRF**	10 (6.7)	0.5	0.529 (0.364–0.636)	0.987 (0.971–1.000)	0.766 (0.556–1.000)	0.963 (0.951–0.972)	7 (4.7)
**COPD**	11 (7.4)	0.4	0.449 (0.286–0.571)	0.953 (0.926–0.978)	0.499 (0.378–0.636)	0.943 (0.928–0.955)	8 (5.4)
**HF**	5 (3.4)	0.3	0.500 (0.500–0.500)	0.971 (0.952–0.986)	0.319 (0.222–0.500)	0.986 (0.986–0.986)	2 (1.3)
**T2D**	4 (2.7)	0.3	0.325 (0.125–0.375)	0.956 (0.943–0.972)	0.297 (0.143–0.375)	0.961 (0.951–0.965)	7 (4.7)

^1^ Percentages are calculated using a denominator of 222 state-wide and 149 metropolitan areas. PPH: potentially preventable hospitalisation; PPV: positive predictive value; NPV: negative predictive value.

**Table 2 ijerph-18-10253-t002:** Validation statistics for the ten best state-wide models for high-risk foot (HRF), with mean odds ratios for predictors in these models.

Variable	Model									
	1	2	3	4	5	6	7	8	9	10
Number of past consecutive years classified as a hotspot	1.57	1.56	1.57	1.56	1.56	1.55	1.64	1.56	1.56	1.36
IER percentile										0.93
IEO percentile		0.99								
Percentage of Aboriginal individuals ^1^	3.43	3.44	3.59	3.33	3.47	6.03	2.55	3.55	3.36	1.20
Percentage of male individuals ^2^	0.71	0.76	0.73	0.71	0.74	0.87		0.75	0.73	0.83
Percentage of individuals aged 75 or above				0.99	0.99					
Weighted distance to nearest ED (km) ^1^		1.05	1.02		1.02		0.74	1.06		
Weighted distance to nearest GP (km) ^1^						0.43				
GP accessibility index percentile								1.00	1.00	1.00
**Validation statistics**										
Sensitivity	0.531	0.504	0.530	0.528	0.520	0.504	0.527	0.517	0.526	0.516
PPV	0.872	0.870	0.869	0.868	0.867	0.865	0.864	0.862	0.861	0.859

^1^ Variable has been transformed. ^2^ Variable has been centred. IER: index of economic resources; IEO: index of education and occupation; ED: emergency department; GP: general practice; PPV: positive predictive value.

**Table 3 ijerph-18-10253-t003:** Presence of predictors in the optimal state-wide and metropolitan models for high-risk foot (HRF), chronic obstructive pulmonary disease (COPD), heart failure (HF), and type II diabetes (T2D).

Variable	Model							
	State-Wide				Metropolitan			
	HRF	COPD	HF	T2D	HRF	COPD	HF	T2D
Number of consecutive past years classified as a hotspot	√	√	√				√	√
IRSAD percentile						√		
IER percentile					√		√	
IEO percentile				√				
Percentage of Aboriginal individuals ^1^	√	√	√	√	√			√
Percentage of male individuals ^2^	√					√		√
Percentage of individuals aged 75 or above		√	√		√		√	
Percentage of individuals aged 75 or above (quadratic term)		√	√		√			
Number of consecutive past years classified as a hotspot: IER percentile ^3^							√	
Weighted distance to nearest ED (km) ^1^				√		√	√	
Weighted distance to nearest GP (km) ^1^		√			√			
GP accessibility index percentile			√					

^1^ Variable has been transformed. ^2^ Variable has been centred. ^3^ Interaction term. IRSAD: index of relative socioeconomic advantage and disadvantage; IER: index of economic resources; IEO: index of education and occupation; ED: emergency department; GP: general practice.

**Table 4 ijerph-18-10253-t004:** Validation performance statistics for the “current hotspots” and “past persistent hotspots” prediction rules.

Model	PPH	Method			
		Current Hotspots		Past Persistent Hotspots	
		Sensitivity	PPV	Sensitivity	PPV
State-wide	HRF	0.684	0.406	0.421	0.8
	COPD	0.667	0.333	0.143	0.6
	HF	0.583	0.28	0.083	0.5
	T2D	0.533	0.286	0.267	0.571
Metropolitan	HRF	0.545	0.273	0.364	0.571
	COPD	0.429	0.24	0.071	0.333
	HF	0.5	0.143	0	NA
	T2D	0.125	0.063	0	0

PPH: potentially preventable hospitalisation; PPV: positive predictive value; HRF: high risk foot; COPD: chronic obstructive pulmonary disease; HF: heart failure; T2D: type 2 diabetes; NA: not applicable.

## Data Availability

Restrictions apply to the availability of health data examined in this study. These data were obtained from the Department of Health, Western Australia and are available from that organization by application.

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
