# Peer review of "Predicting Future Geographic Hotspots of Potentially Preventable Hospitalisations Using All Subset Model Selection and Repeated K-Fold Cross-Validation"

_ijerph, 2021, doi:10.3390/ijerph181910253_

Round 1
Reviewer 1 Report
Overall I think that this manuscript reads well and applies understood methods to the detection of hotspots for PPH. These methods are not particularly novel, and more frequently used methods are dismissed without discussion. I.e., why did the authors choose the methods they did over others?
There has been a large quantity of work in this area that is not discussed or differentiated from in the current manuscript. The review of literature on this topic seems out of date, and missing some relevant manuscripts. particularly as this type of study has seen renewed interest in light of the Covid pandemic.
Reviewer 2 Report
The article presents an interesting topic on the geographic hotspots of preventable hospitalisations using optimal prediction models. The risks factors and health behaviour of the population concerned are important determinants that are commonly unknown or even limited.
-The summary is written clearly and concisely according to the required academic standards.
-The introduction provides a valuable background to the subject.
-The discussion section provides evidence for the issue addressed as well as the limitation of the study.
Overall, it is an interesting and well-written paper.
Some suggestions the authors may consider improving the clarity of the paper:
-The paper's objectives are clear, but more arguments about the need for the study could strengthen the paper.
-The advantages of the optimal prediction models compared to other existing methods should be more clearly outlined.
-Some additional information about the hospital catchment areas limitation would be necessary.
-Enumeration of all diagnosis codes, not necessary Line 272-281-
-The fig.1.: it is unnecessary to present the small-area statistical boundaries for the UK and US; maps of MSOAs across England and Wales and CBGs across the US 88 state of Illinois (as examples) could be confusing here and should be removed (fig 1c,d)
-Due to the spatial heterogeneity of the small-area statistical boundaries, more critical insights of applied methods should be addressed. For example, the statistical boundaries are artificially constructed, and the scale effect is present. The problem of MAUP could be more detailed in that case.
-The conclusion section could be more extended to better support the results.
